



# The geomagnetic data of the Clementinum observatory in Prague since 1839

Pavel Hejda[1], Fridrich Valach[2], and Miloš Revallo[3]

[1]Institute of Geophysics, Academy of Sciences of the Czech Republic, Boční II/1401, 141 00 Prague, Czech Republic
[2]Geomagnetic Observatory, Earth Science Institute, Slovak Academy of Sciences, Komárňanská 108, 947 01 Hurbanovo, Slovakia
[3]Earth Science Institute, Slovak Academy of Sciences, Dúbravská cesta 9, 840 05 Bratislava, Slovakia

**Correspondence:** P. Hejda (ph@ig.cas.sk)

**Abstract.** The historical magnetic observatory Clementinum operated in Prague from 1839 to 1926. The data from the yearbooks that recorded the observations at Clementinum have recently been digitized and were subsequently converted in this work into the physical units of the International System of Units (SI). Introducing a database of geomagnetic data from this historical source is a part of our paper. Some controversial data are also analysed here. In the original historical sources, we

identified an error in using the physical units. It was probably introduced by the then observers with the determination of the temperature coefficient of the bifilar apparatus. By recalculating the values in the records, some missing values are added; for instance, the temperature coefficients for the bifilar magnetometer, the baselines, and the annual averages for the horizontal intensity in the first years of observations were re-determined. The values of absolute measurements of the declination in 1852, which could not be found in the original sources, were also estimated. The main contribution of this article rests in critically

reviewed information about the magnetic observations in Prague, which is as complete as no other so far. The work also contributes to the space weather topic by revealing a record of the now almost forgotten magnetic disturbance of 3 September 1839.

## 1 Introduction

Reliable long time series of geomagnetic records are needed for studying long-term behaviour of the Earth's magnetic field, for instance its secular variation (Cafarella et al., 1992). At present, high resolution direct geomagnetic measurements provide extremely useful information about changes of the geomagnetic field. Magnetic and auroral records are available from a large number of stations situated at auroral as well as lower latitudes and are supplied with satellite data and solar observations. However, the time span of these high resolution data is limited to past few decades. On the other hand, paleomagnetism

provides low resolution data on the field over timescales of thousands of years to hundreds of millennia. To reconstruct the secular variation it is desirable to have high-quality measurements of the geomagnetic field covering as long a time span as





possible. Historical geomagnetic records can be useful to deal with this task, however they can be heterogeneous, incomplete or consist of corrupted data. Their processing should involve several steps like digitization, conversion from scale units to proper physical units, dealing with inaccuracies or missing data, searching for accompanying information like auroral records, etc. To obtain complete datasets, compiling archive records from several magnetic observatories is needed. In previous works by e.g. Jackson et al. (2000); Jonkers et al. (2003); Arneitz et al. (2017), historical geomagnetic data were collected, processed and analysed with the aim of preparing datasets usable for broader geoscience community.

Historical geomagnetic observations are thus understood to be an important source of information about the structure and time behaviour of the geomagnetic field. The magnetic needle has already been commonly used in naval navigation since the 13th century, however at first it was supposed that the needle was oriented towards the true north. The fact that the orientation of magnetic needle depends on the position was documented in the 16th century by explorers sailed in Atlantic and Indian Oceans. On the other hand, the first sustained series of measurements at a single site in Greenwich showed that the geomagnetic field was subject to time-dependent change. The magnetic needle moved of about 7 degrees westerly over period 1580–1634. The first magnetic inclinometer (a magnetized needle rotating on a horizontal axis in the vertical plane of the magnetic meridian) was constructed by London compass maker Robert Norman at about 1580.

Relative magnetic intensity data began to be compiled from 1790s by comparing the time it took a magnetic needle displaced from its preferred orientation to return to it, or by the duration of a given number of such oscillations.

The method of absolute determination of magnetic intensity was developed by Carl Friedrich Gauss in cooperation with Wilhelm Weber in 1832. The method combines vibration and deflection experiments in order to separate intensity of the magnetic field and magnetic moment of the magnet used in the experiment (Gauss, 1833). In 1833, Gauss and Weber finished the construction of the Magnetic Observatory of Göttingen and developed or improved instruments to measure the magnetic field, such as the unifilar and bifilar magnetometers. The Göttingen Observatory became the prototype for many other observatories worldwide.

Improvement of observatory practice was not a goal of Gauss's work, but just a tool for understanding the nature of the Earth magnetic field. Gauss and Weber therefore joined the activity of Alexander von Humboldt in establishing a worldwide network of observatories, known as the Göttingen Magnetic Union (GMU), that made simultaneous measurements at specific intervals ("term days"). The coordinated measurements started with 9 European observatories (6 of them in Germany) in 1936 and the number increased to 31 observatories in 1841.

Prague Observatory joined the GMU in 1839. The observatory had its seat in Clementinum College situated in the Old Town close to Charles Bridge. The College was established by Jesuits in 1566. Since 1622 Jesuits administered Charles University and transferred the University Library to Clementinum. In the beginning of the 18th century the Astronomical Tower was build there and in 1752 the Astronomical Observatory was established. The first director, Joseph Stepling, started soon also meteorological measurements. An uninterrupted series of high quality temperature measurements dates back to 1 January 1775 and is well known to climatologists all around the world. Thanks to Karl Kreil, the importance of magnetic observations does not lag behind the meteorological ones.





Kreil came to Prague Observatory from Milano in 1838. In Milano, he was visited by Gauss's assistants and became enthused by this new research discipline. After he was informed about his transfer to Prague, he took care of acquisition of magnetic instrumentation. The equipment of the observatory corresponded to the prototypes used in Göttingen. The variation observations were installed in a large corridor of the Astronomical Observatory. The building was not completely iron-free,

however all iron objects were removed from the vicinity of the instruments and test observations did not indicate any magnetic contamination. The instruments were arranged in such a way that they did not interfere and that one observer was able to perform the eye-observations of declination, horizontal intensity and inclination in time intervals of 5 minutes during the "term days" or even more frequent observations of declination and horizontal intensity during the periods of disturbed magnetic field. Compared with modern observatory variometers, the instruments were quite massive. The weight of the declination "needle"

(magnetic rod in the form of parallelepiped) was 1682 gram and the weight of rod in the bifilar magnetometer was 2780 gram. The magnetic rods in bifilar magnetometers constructed by the famous instrument makers M. Meyerstein in Göttingen and advertised in the Resultate aus den Beobachtungen des magnetischen Vereins im Jahre 1837 weighed even up to 25 pounds (nearly 12 kg).

The absolute measurements were carried out in the Imperial gardens near the Prague castle, in a place sufficiently distant

from any buildings. As there was no hut or shelter at the beginning, the measurements were carried out in shine windless days in order to eliminate the influence of wind on the rod oscillations.

Regular magnetic observations were started in July 1839. In the first year the observations were performed 17 times per day, however, their frequency soon decreased to 10 measurements and later to 5 and 3. Simultaneous measurements at specific intervals (term days) in the frame of the Göttingen Magnetic Union were performed up to 1849. In the first decade also

measurements with the frequency of 2 minutes were carried out during periods of magnetic storms.

It was important that from the very beginning all measurements were published in the yearbooks called Magnetische und meteorologische Beobachtungen zu Prag. It was the best way how to save all data without a danger of lost by fire, flooding or incompetency and disregard of the future staff. The yearbooks contain tables of variation observations (magnetic and meteorological), reports on absolute magnetic measurements and discussion to their conversion to physical units. In the first volumes

also observations of vegetation were included (development of sprouts, leafs, flowers, etc.).

In 1850 Kreil left Prague for Vienna where he established the Central Institute for Magnetic and Meteorological Observations in Austria. Prague observatory was operating until the beginning of the 20th century. Due to the increasing urban noise, the observations were finally limited to the declination in 1905 and the observatory was closed in 1926.

Most observatories operating within the Göttingen Magnetic Union were closed already in the 1840's or 1850's. Just a

few observatories established before 1850 were in operation up to the year 1900 or later. According to the information about observatories from the regional reports in (Gubbins and Herrero-Bervera, 2007) and from the list of observatory yearly means in the WDCs, these were Clausthal, Colaba, Greenwich, Göttingen, Helsinki, Kew (London), Milano, Munich, Oslo (Christiania), Prague, Ekaterinburg (Sverdlovsk), Toronto and Wien.

Although summary data of the Prague observatory have been used for research tasks already by Wolf and his successor

Wolfer for calibration of sunspot numbers, the data as a whole have up to now stayed available only in the printed form of





yearbooks. The recent interest in historical data led us to the decision to digitize the data. At the first stage, all volumes of the yearbooks were scanned and transferred into pdf files. The complete collection of the scanned original yearbooks of the Clementinum/Prague magnetic observatory are made available via the web page of the Institute of Geophysics of the Czech Academy of Sciences (https://www.ig.cas.cz/en/prague-observatory-yearbooks/).

Although the OCR was part of the scanning process, the text files created by this procedure contained too many errors to be usable for data digitization. The manual digitization was thus carried out by means of spreadsheets with pre-programmed templates that allowed also for preliminary data check and repair of rough errors: computed monthly means were compared with monthly means published in the yearbooks. All declination and horizontal intensity data of regular observations have been already digitized. The digitization of data from disturbed periods will follow.

As the above-mentioned calibration of sunspot numbers by Wolf and Wolfer shows, geomagnetic observatory data can serve as a proxy of space weather parameters. Old geomagnetic records can thus provide invaluable information for space weather studies long time before the launch of artificial satellites. It is known that the most severe geomagnetic disturbances, including the well-known Carrington event in 1859, occurred during the historical solar cycles. Searching for geomagnetic disturbances in the past and their analysis is important for understanding the causers of extreme geomagnetic activity. At present, this kind of
research can be beneficial for setting up reliable space weather forecast models. It is known that besides of the the ring-current storms there are even more violent geomagnetic field variations called auroral substorms. In (Valach et al., 2019), the data from Clementinum observatory were used to study strong geomagnetic disturbances which were interpreted as substorms.

   Structure of the paper is as follows. In Section 2, we present the database consisting of the Clementinum data. In Section 3, we deal with a-few-year period of corrupted declination record. The key part of the paper rests in Section 4 which is devoted to
the study of horizontal intensity registered at Clementinum using a bifilar magnetometer. An example of interesting magnetic disturbance is presented in Section 5.

## 2   Database of the Clementinum magnetic data

The magnetic data published in the yearbooks Magnetische und Meteorologische Beobachtungen zu Prag include absolute observations, regular variation observations, more frequent observations during magnetically disturbed conditions and simulta-
neous observations on term-days agreed in the frame of Göttingen Magnetic Union. The first four volumes cover the first four subsequent periods: from July 1839 to July 1840, from August 1840 to July 1841, from August 1841 to July 1842, and from August 1842 to December 1843. All other volumes coincide with the calendar years.

### 2.1   Absolute observations

As the Clementinum building, where the daily variation observations were carried out, was not situated in a completely iron-
free environment, the absolute observations were performed in the Emperor Garden near the Prague Castle. At the beginning, the observations were held in the open air, later a wooden hut was built. Declination, inclination and horizontal intensity were measured. The latter one followed the procedure developed by Gauss (1833). The absolute measurements started as late as





in August 1840. In the first decade, the observations were performed sporadically and their number has been increasing up to monthly frequency. In 1860, the absolute observations were moved to a former chapel in Seminar garden on the uphill of

Petřín (called as Laurenziberg in the yearbooks). The absolute measurements were compared with variation measurements to estimate the base value and size of the scale unit.

## 2.2    Regular variation observations

The variation magnetometers were installed in a 4.5 m wide corridor below the Astronomical Tower of Clementinum. The instruments including telescopes for scale readings were spaced out in such a way that one observer was able to carry out

measurements of declination, horizontal intensity and inclination within 2 minutes. Regular variation observations started in July 1839 by measurements of declination and horizontal intensity. Two months later, measurements of inclination and oscillation period of inclination needle were added, the latter providing some information about the total field. In the first year, 19 observations were carried out per day and later the numbers of observations were gradually reduced.

Time stamps in the yearbooks show Göttingen astronomical time. Compared to Prague astronomical time, the difference

is 18 minutes. According to astronomical convention, 0 hour means noon and 12 hours midnight. More precisely, declination is measured on the hour, horizontal intensity 2 minutes later and inclination again 2 minutes later. The oscillation period of inclination needle is observed before and after the above mentioned measurements. The summary of magnetic variations is given in Table 1. The time of measurements in Table 1 corresponds to the Göttingen "civic time", i.e. 12:00 corresponds roughly to 11:20 UT.

As the magnetization of the needle is temperature dependent, the temperature inside the case with needle was also recorded, at the beginning twice a day, later during all measurements. Kreil was aware that the temperature dependence of the needle as well as the issues of the geomagnetic observations in general were not yet sufficiently understood and that is why the raw data in scale units were published. The data in physical units were published from Vol. 33 and publication of scale unit data was stopped in Vol. 45. We have decided to digitize first the declination and horizontal intensity data. Transformation of the

data available only in scale units to physical units will be reported in the next sections of this paper. The complete data set of declination from 1839 to 1917 and horizontal intensity from 1839 to 1904 is provided in Supplement.

## 2.3    Records of magnetic disturbances

If the observers on duty noticed exceptionally fast change of declination or horizontal intensity, they started observations of these components with higher frequency. The interval between measurements was in range from 24 seconds to 10 minutes with

a typical period of 2 minutes. About 150 events were recorded from September 1839 to December 1843, which represent 200 full pages in the yearbooks. In the preface to volume 5, Kreil assessed hitherto known practice and came to the conclusion that only sufficiently large disturbances are worth to be included into the reports in the future. These high-frequency measurements were stopped in 1851 after Kreil left Prague to establish the Central Institute for Magnetic and Meteorological Observations in Vienna. However, the information about disturbed days was not fully omitted. Some brief reports on magnetic storms

were published in monthly or yearly summaries. The yearbooks include also observations of northern lights. They are scattered



between above mentioned lists of disturbed days and observations of atmospheric phenomena. About 20 observations in Prague between 1839 and 1905 were recorded. Thanks to the dark night sky in the 19th century, the polar light was observable even in the city downtown. Information on the polar lights appearance across Europe and North America was also published.

### 2.4 Term-days observations

As mentioned in the Introduction, Prague observatory shortly joined the simultaneous observations of the magnetic field on selected term-days organized by Göttingen Magnetic Union (GMU). Four term-days per year were agreed from 1839, started on the Friday preceding the last Saturday of the month, at 10 p.m. Göttingen mean time, in February, May, August and November. The observations were carried out with 5-minute steps for 24 hours. The results were published in six volumes of the yearbooks Resultate aus den Beobachtungen des Magnetischen Verein (Gauss and Weber, 1837–1841,1843). The publication

of Resultate ceased in 1843, after Weber joined the political group the Göttingen Seven, which protested against constitutional violations of King Ernest August of Hannover, and had to leave Göttingen. Out of GMU, Edward Sabine, who coordinated magnetic observatories built up by the Great Britain Government and East Indian Company, proposed that eight additional simultaneous observations be performed on the Wednesday nearest the 21th day of the eight remaining months, the hour of commencement being the same as in GMU (Kreil, 1840b, p. 136–138). Kreil supported the proposal and carried out the

term-days measurements every month. He continued doing so until the year 1849.

### 3 Corrupted data of magnetic declination around the year 1852

Figure 1 with the time series of magnetic declination clearly shows the secular variation (a systematic trend in which the orientation of the horizontal projection of the geomagnetic field vector shifts to the east) as well as a distinct seasonal variation (deviations resembling sinusoids). However, in the period that began sometime in early 1852 and lasted for about two years, the

course of the declination had a different character than during the rest of the period. This special period begins with a sudden decrease in the size of the declination by about 20 angular minutes; subsequently the time series continues, as if without secular variation. Such a course of the magnetic element is unexpected and highly probable is also incorrect.

Indeed, the yearbook for 1852 (Böhm and Kuneš, 1855, p. IV) also paid attention to this peculiarity. It presents comparisons of the absolute value determined by absolute measurements with the values determined from the variation apparatus using the

formula from the previous year (i.e. from 1851). These comparisons show that the difference between these values has been steadily increasing almost systematically; in June 1852 the difference ranged from 4 to 5 angular minutes (with a negative sign), in December it was already more than 12 angular minutes (again with the minus sign). Böhm and Kuneš (1855, p. IV) wrote: "Over the next year, this difference widened without being able to determine its cause, and only later did it become apparent that a spider, which was in the box where the small magnet of the variation apparatus was hung, caused these differences."

The variations in the yearbook in 1852 and in the first part of 1853 were thus contaminated, likely by a spider web which in some way mechanically affected the position of the magnet in the device. There is probably no way to remedy these variations, because the spider could have rebuilt its web in a way we have no way to determine. Therefore, we cannot reliably determine





the annual averages for the years 1852 and 1853 from those variations. After examining the table comparing the absolute measurements made in June and October 1853 with the absolute data determined from the variations (Böhm and Karlinski,

1856a, p. III), it seems that for June and October 1853 there are no more systematic deviations between absolute measurements and variations; although there are some differences, but they do not seem to be systematic but rather random, with different signs. Therefore, we assume that the variations in the second half of 1853 (including June) are already correct.

However, despite the contaminated variation data, we could still obtain some substitute values for the annual averages of absolute declination for 1852 and 1853, which could contain valuable information; more valuable than mere interpolation from

existing surrounding annual averages.

Although the absolute measurements of 1852 were not found in contemporary sources, there are available the records of differences between the absolute values from variations and from direct measurements (Böhm and Kuneš, 1855, p. IV), those which we have already mentioned above. These records were captured on the following days: 22 and 23 June, 6 and 7 July, 21 and 22 October, and 24 December of 1852. Using the formula for converting declination variations to absolute declination and

the coefficients which are valid for 1851 together with the records of those calibration measurements, we may easily reconstruct the results of the absolute measurements.

The absolute measurements of declination in June and early July can be interpreted as measured during the summer solstice, the measurements in October are closest to the autumn equinox, and the December measurements are almost exactly at the time of the winter solstice. Figure 2 compares the absolute values in these three time points to the monthly averages in the

surrounding periods. We admit that the comparison is not entirely perfect, because we have compared the averages of absolute measurements with the monthly averages at 10.00 pm (they should be the least affected by daily variation), but the absolute measurements were made between 8.00 am and 10.00 am (local solar time). Even so, we can conclude that the values fit well into the time series. This, to some extent, justifies us to declare the average of the declination calculated from these three instances to be a substitute for the annual average of the declination in 1852. It gives 14° 17.29'. Note that declination at that

time was western in Prague. In accordance with the convention in which the positive direction for the declination is eastern, a negative sign should be added to the then value of declination in Prague.

A substitute value for the annual average of 1853 can be calculated from the monthly averages from June to December 1853, which can be found in the yearbook (Böhm and Karlinski, 1856a). The formula obtained from the measurements in the following year (Böhm and Karlinski, 1856b, p. III) can be used for these calculations (Böhm and Karlinski, 1856b, p. III). The

resulting absolute value of the western declination for this year is thus 14° 10.49'.

The database on the website of the Institute of Geophysics of the Czech Academy of Sciences also preserves the original variation data of magnetic declination affected by the spider net. Being aware of the limitations of these data, they can still be considered useful for studying rapid variations such as magnetic storms, at least for the purpose of qualitative analysis.



## 4   Horizontal intensity in first years of the Clementinum observations

Measurements of horizontal intensity were a relatively new issue at the time of the beginnings of geomagnetic observations in the Prague Clementinum.The methodology of these measurements was invented only a few years before, the absolute method in 1832 and the method for the variations observations in 1837 (e.g., Garland, 1979). Probably the novelty of this issue caused some ambiguities or imperfections in the first records of these measurements in the Clementinum yearbooks. In this section we describe some of the problems we encountered while studying the oldest Prague records during the preparation of this article.

Solving these problems requires a basic knowledge of the operation of a bifilar magnetometer, which is clarified below.

### 4.1   The equation of the bifilar magnetometer

The bifilar magnetometer, the type of an instrument that was used for to perform the observations of the horizontal intensity variations in the mid 19th century (Gauss, 1838), has recently been reminded to the geomagnetic community by Garland (1979) and Nevanlinna (1997).

The main part of the instrument was a large magnetic needle that was suspended by two long fibres (a typical length of the fibres was more than a metre). The fibres maintained the horizontal position of the needle and at the same time allowed the needle to turn in the horizontal plane; as they were very thin and hanged close to each other (a typical distance between them was a few of centimetres), they put only a small resistance against the rotary horizontal movement. The length of the needle was about one meter and its typical weight was of the order of two kilograms. The console from which the fibres hanged could

be revolved around the vertical axis (Fig. 3).

During the initial adjustment of the instrument, the console was revolved so that the magnetic needle came to a position perpendicular to the magnetic meridian. The console was locked in this position.

When there was a small change of the magnitude of the magnetic force, which balanced the torque caused by the torsion of the pair of fibres, the magnetic needle slightly deviated from the perpendicular direction in the horizontal plane. In a small

mirror, placed on a magnetic needle at the axis of rotation, the reflection of a scale was observed with a telescope with an index line in its objective. It was a scale that was fixed and was placed, for example, on the wall of the room. This made it possible to determine by which angle the needle deviated (Fig. 4).

The change in magnetic force could be due to either a change in horizontal intensity or a change in needle magnetization due to a temperature variation. It can be shown (e.g. Nevanlinna, 1997) that the balance between the torsion of a pair of fibres

and the rotational effect of a magnetic force yields the equation

$$\frac{\Delta H}{H} = c_1(h - h_0) + c_2(t - t_0), \tag{1}$$

where $H$ is the absolute value of the horizontal intensity, $h_0$ is the position of the index line observed through the telescope on the scale at the time of the instrument adjustment, and $t_0$ is the temperature of the magnetic needle at the time of the instrument adjustment. At some later time after the instrument is adjusted, when an observation is being carried out, the position of the

index line in the telescope is $h$, the temperature of the needle is $t$, and the value of the horizontal intensity differs from the value





during the adjustment by $\Delta H$. Equation (1) makes it possible to observe the variations in horizontal intensity by monitoring the position of the index line on the scale and recording the temperature — thus, the bifilar apparatus is a variation device. However, in order to be able to use such a variation device, it is necessary to know the scaling coefficient $c_1$ and the temperature coefficient $c_2$.

In the yearbooks from the Clementinum Observatory in Prague, the values of constants $c_1$ and $c_2$ have been explicitly given together with tables containing the observed index line positions (in scale divisions) and the observed temperatures, but only since 1855. In older yearbooks, only the scaling coefficient was explicitly stated. Although the yearbook (Böhm and Karlinski, 1857, p. XV) contains a record of the calculation of the temperature coefficient for the previous period, in our opinion this calculation is erroneous, and moreover it is valid only after the beginning of 1846, when the older bifilar apparatus was

replaced by another apparatus. Therefore, two partial problems we had to solve before the equation (1) could be used:

–   determining the temperature coefficient for the older bifilar apparatus (1/1/1840 – 31/12/1845),

–   revealing the error or checking the calculation of the temperature coefficient for the newer bifilar apparatus (since 1/1/1846).

In the following section (Sec. 4.2) we will report on the solution of these two tasks.

**4.2    Determining the temperature coefficient for the oldest observations of the component $H$ variations**

To determine the temperature coefficient $c_2$ for the initial observations of the variations of the horizontal intensity, we can use time series of two quantities, which have been recorded in the old yearbooks: variations in horizontal intensity expressed in divisions of scale $h$ and temperature $t$ expressed in degrees Réaumur (abbr. °R; 1°R = 1.25°C). Temperature $t$ was measured near the magnetometer and represented the temperature of the magnetic needle. The scaling coefficient $c_1$ has been provided

in the yearbooks and we consider its values to be correct.

A similar task, which is to find the temperature coefficient from values of the quantities $h$ and $t$, was solved for observations at the Helsinki Observatory by Nevanlinna (1997). However, in the case of Helsinki, it was typical that the temperature variations during the day were very large. It was probably due to that their observation room was apparently unheated and insufficiently insulated from the outside environment. This allowed Nevanlinna (1997) to consider the geomagnetic variations during the

magnetically quiet days of the winter season to be negligibly small compared to the effect caused by the changes in needle magnetization due to the daily temperature variations. If during winter quiet days the changes of $h$ might be considered only as a consequence of temperature variations, linear regression worked well to estimate the temperature coefficient.

However, in the case of the Clementinum Observatory, the situation is different. The thick wall of the building where the observations were performed ensured that the diurnal variation of the temperature of the magnetic needle was almost

completely smoothed out. Only a seasonal variation with one-year period was observed in the series of temperature inside the observation room.

Because we know that the variations in $h$ caused by the change in needle magnetization were much greater than the variations due to the actual change in magnetic field, we might use a similar approximation as Nevanlinna (1997) on a time scale of several





months to years. Also in our approximation, we considered changes in $h$ only as a consequence of changes in temperature $t$, but
unlike (Nevanlinna, 1997) they were not the daily variations of the values of $h$ and $t$, but the seasonal variations. This allowed
us to use linear regression in the same way to estimate the temperature coefficient.

We also used another simplification, namely we considered only the temperature dependence for the magnetization of the
needle. Like Nevanlinna (1997), we did not consider that the magnetization of the rod at any fixed specific temperature could
change over time during the studied period. Although the yearbook (Böhm and Karlinski, 1857, p. XV) stated that the mag-
netism of the rod (needle) "weakened" over time, we think it was an interpretation based on incorrect data. According to our
reasoning, the authors of the yearbook confused the physical units in the data on which they based their interpretation. In our
opinion, the correct data on the temperature coefficient should be as stated in Table 2. In the yearbook (Böhm and Karlinski,
1857), the authors claimed that they had the absolute magnetic unit per degree of Réaumur. However, based on our analysis in
Section 4.2, we claim that the figures for all three years mentioned (1846, 1849 and 1856) should have been correctly reported
in parts of the whole horizontal intensity per degree of Réaumur.[1]

The values in Table 2 do not indicate any systematic increase or systematic decrease in the $c_2$ coefficient; we think that
the reason why the listed values differ from each other is rather the inaccuracy with which those values were determined.
Therefore, we prefer to assume in our study that the $c_2$ coefficient, which stems from the temperature dependence of the needle
magnetization, was still approximately the same value for the same needle.

However, from information in the yearbooks, it can be concluded that two needles or two bifilar devices were used. The first
bifilar apparatus was in operation from the beginning of the magnetic measurements in Clementinum until 31 December 1845.
On 1 January 1846, another apparatus, which was working with another magnetic needle, started to be employed. It means that
two values of the $c_2$ coefficient have to be determined: the first of them being valid until the end of 1845 and the second one
being valid from the beginning of 1846.

Regarding the first case (i.e., for years 1840-1845), we found no specific mention about the value of coefficient $c_2$ in the
yearbooks. Therefore, in Section 4.2.1, we calculate it as a new and hitherto unknown numerical value. In the second case, for
the period from the beginning of January 1846, three values of the coefficient were given in the yearbooks (we have listed them
in Table 2), but we need to dispel doubts about the correctness of these units. The calculations in Section 4.2.2 thus serve to
verify our assumption of correct units, rather than to determine some other numerical value.

### 310    4.2.1    Calculating the temperature coefficient for the bifilar apparatus until 31 December 1845

We estimated the value of the temperature coefficient for the period from 1 January 1840 to 31 December 1845 for six isolated
time periods, between which, according to notes in the yearbooks, the setting of the bifilar apparatus was changed. The re-
adjustment of the apparatus was probably performed for two reasons:

– either the fibres on which the needle has been suspended have ruptured,

---

[1]In some places, there are errors in the yearbook when the value in parts of the whole horizontal intensity is given instead of the value in absolute units.
The error can be eliminated by multiplying by 1.9, which is the value of the then horizontal intensity in absolute units of the system mm-mg-s introduced by
C. F. Gauss.





– or the instrument had to be re-positioned due to the secular variation, as a result of which the deviation of the magnetic
       needle systematically increased to the limit of the measuring range of the instrument.

Using coefficient $c_1$, the values of which we read from the yearbooks, we converted the variations given in the scale divisions
to the numerical values given in the parts of the whole horizontal intensity.

       Subsequently, we calculated temperature coefficients for individual periods using the method of least squares. In doing so,
we determined the parameters of the function

$$\frac{\Delta H}{H} = -c_2 t + c_3 \,. \tag{2}$$

In this linear equation, $c_2$ is the temperature coefficient sought and $c_3$ is the parameter, which determines the value of the
fraction $\Delta H/H$ at the temperature $t = 0°\mathrm{R}$ (the specific value of $c_3$ is of no interest for this study). The obtained values of the
temperature coefficient for the period 1840-1845 are given in Table 3. According to these data, the mean of the temperature
coefficient is

$$c_{2,1840-1845} = (0.000\,574 \pm 0.000\,080)H \qquad (\text{in}\,°\mathrm{R}^{-1})\,. \tag{3}$$

Since we do not have another estimate of the temperature coefficient for the very first years of observations in Clementinum,
we will use this numerical value in further calculations (Sect. 4.3).

### 4.2.2 Verification of the temperature coefficient of the bifilar apparatus from 1846 to 1854/55

Even from 1 January 1846, when a new bifilar apparatus began to be used for observations of horizontal intensity, until the
end of 1854, the values of the temperature coefficient were not given in the yearbooks. In the 1855 yearbook, the temperature
coefficients were back-calculated, but due to uncertainties about the physical units in which their results were given, we
considered it necessary to verify them with our own calculations. We proceeded in the same way as in Section 4.2.1.

       For the period from the beginning of 1846 to 8 March 1855 we had five continuous time periods at our disposal, for which
we determined the temperature coefficient satisfying equation (2) by the method of least squares. The partial values are listed
in Table 4.

       The mean value for the temperature coefficient calculated from the data in Table 4 is

$$c_{2,1846-1854} = (0.001\,186 \pm 0.000\,168)H \qquad (\text{in}\,°\mathrm{R}^{-1})\,. \tag{4}$$

       The value that we obtained is in good agreement with the corrected data stored in Table 2. After this verification, the
numerical values in Table 2, or rather their mean value, may be taken as the temperature coefficient for the whole period from
1 January 1846 to 31 December 1854:

$$c_{2,1846-1854} = \frac{0.001\,172 + 0.001\,273 + 0.001\,108}{3} H = 0.001\,184 H \qquad (\text{in}\,°\mathrm{R}^{-1})\,. \tag{5}$$

       We will use this value in Section 4.3 in our calculations for the period up to the time when the values of the temperature
coefficient starts to be provided in the yearbooks (in the headers of tables that contain the variations given in scale divisions).





### 4.3 A baseline for the horizontal intensity in years 1840-1854 and the annual means

At the beginning of the geomagnetic observations at Clementinum, two types of measurements were performed:

- occasional measurements of the whole horizontal intensity by means of the Gauss absolute method,

- regular observations of the variations of the horizontal intensity (expressed in parts of the whole horizontal intensity).

These two types of measurements were linked only by the fact that observers needed a value of the total horizontal intensity to express variations in horizontal intensity in absolute units. However, in contrast to nowadays practice, the baselines of the variation apparatus were not calculated afterwards. Therefore, in this section we will calculate the baseline for the period from 1840 to 1854. Since 1855, the yearbooks provide the baseline values (under the designation "Constante").

To calculate the values of the baseline, we followed the procedure that became the standard method today, i.e. we compared the trends in the time series of registered variations of horizontal intensity (only the pure variations with no baseline added) with the values of horizontal intensity determined by absolute measurements. The time series of the registered variations with no baseline added were calculated for each of the continual periods separately (Tables 3 and 4). Using the tabulated variations, which are provided by the yearbooks in scale divisions, together with scaling coefficient $c_1$ and temperature coefficient $c_2$, we obtained the values for the variations of the horizontal intensity expressed in the absolute units.

Figure 5 in panel (a) shows that there are some instances in the time series when the values change abruptly. These are the discontinuities due to re-adjustments of the bifilar device. In one case, namely between 30 December 1845 and 1 January 1846, the abrupt change was caused by replacing the old bifilar instrument by a new one. In the next step, we therefore treated the discontinuities in the time series. We did it by moving manually the entire continuous sections of the time series in the vertical direction so that we connected the ends of the previous sections with the beginnings of the following sections. The resulting continuous time series is shown in Fig. 5, panel (b), where the values are given in the absolute units of Gauss's original unit system that was based on millimetre, milligram, and second. By multiplying those numerical values by a factor of $10\,000$, the original unit can be changed to modern nanotesla unit.

Other necessary data are the results of absolute measurements of the horizontal intensity. Figure 6 displays the data on the absolute measurements that we found in the yearbooks. A comparison with the course of the data of the Munich Observatory, which is only 300 km away from Prague, shows at first sight that the absolute measurements before 1844 deviate from what we would expect. A confrontation with the IGRF model also shows that the absolute measurements in the first years of geomagnetic observations in Prague were probably characterized by a disproportionately large error, likely systematic observational error. Therefore, for this period, we preferred to use the value of horizontal intensity, which we obtained by extrapolation based on a linear regression from data between 1844 and 1853.

The final baseline that we then obtained is presented in panel (c) of Fig. 5. We also need to mention that to determine the baseline we used the variation data that were observed at 04.00 pm, because absolute measurements in the first years of the observatory's operation were usually performed around 04.00 pm, which is information found in the yearbooks.





By adding the baseline values to the time series of the variations, we obtained time series of the absolute values of the horizontal intensity (Fig. 5, panel (d)). From these data, we then calculated annual means of the horizontal intensity for Prague (see data in Table 5). The comparison of the annual means that we obtained (Fig. 6) with the annual averages from Munich

indicates a good agreement in the trend of the time series at these two not-too-far-away observatories. We believe that this good agreement can be considered a confirmation of the correctness of the calculations we made in Sections 4.2 and 4.3.

## 5    An example of recorded strong magnetic disturbances: the event on 3 September 1839

In addition to obtaining the time series that capture the secular variations of the geomagnetic field, the study of historical records from old observatories has another, at least equally important, benefit. This benefit is the records of intense magnetic

storms that occurred in the 19th century. As an example of such a record, we show here the course of the horizontal intensity during an interesting magnetic storm, which commenced in the evening of 3 September 1839. Subsequently, after midnight and in the early morning time, two sudden, relatively deep, and short-lasting drops in horizontal intensity appeared in the Prague records. Probably these two swift variations were local phenomena, probably substorms or some other variations that were caused by electric currents in the auroral oval. Here we assume that the auroral oval was reaching as far as Central Europe

at that time. Our assumption that the auroral oval was located in relatively low geographical (or magnetic) latitudes around 3 September 1839 is based on observations recorded in a little-known document (Snow, 1842) from the first half of the 19th century.

Robert Snow, who observed the Northern Lights from September 1834 to September 1839, reported about the event which occurred on 3 September 1839. He claimed that this was the second most beautiful aurora borealis he had ever observed (Snow

1842, p. 15). He made observation of this event at Ashurst in West Sussex, England, this place having geographical coordinates 51°16′N and 0°1′10″W. The geographical latitudes of Prague and Ashurst differ by 14.4° (Table 6). The difference between local times of these two places is thus about one hour. Although this particular geomagnetic disturbance and the accompanying aurora borealis are not well known to the scientific community today, Snow (1842) states that the press at the time informed about this aurora.

The aurora borealis accompanying the magnetic storm of 3 and 4 September 1839 was also observed in Prague. However, Kreil (1840b) writes that they could not pay full attention to the observation of this phenomenon because they measured magnetic field, namely declination and horizontal intensity, all the time.

Other examples of interesting geomagnetic disturbances observed in the Clementinum are the extreme variations of the magnetic field that occurred on 17 November 1848 and 4 February 1872 (Valach et al., 2019). We believe that the digitized

records from the Clementinum observatory of Prague will provide an opportunity to study other interesting events.





# 6 Conclusions

The presented study can be primarily conceived as a presentation of the unique database collected from geomagnetic registration at the Clementinum Observatory in Prague during the 19th century. From another point of view, this study can serve as an example of how to deal with historical geomagnetic data in context with current research trends in geomagnetism. It is also
documented here, that putting the records in proper physical units and coping with data inconsistencies is an essential part of the research.

In dealing with proper scaling of the historical geomagnetic records, this study partly builds up on earlier work by Nevanlinna (1997). The major outcome of these efforts is the proper adjustment of thermal coefficient in the equation for bifilar magnetometer. Having this constant determined, it is possible to re-scale the data in geomagnetic records and to reconstruct
the time series for horizontal intensity. As such, the adopted geomagnetic records can be cast in consistent forms to serve as a reliable source of information for research purposes. As detailed in the main part of this paper, the procedure of computing the temperature dependence here in some sense can be considered as complementary to that in (Nevanlinna, 1997).

Closer study of the early Clementinum data reveals an interesting magnetic disturbance which is nearly unknown and almost forgotten among the present-day geomagnetic community. There is an indication that this disturbance can be interpreted as
caused by currents related to the auroral oval transiently extending to mid-latitudes. We proposed that this disturbance could be identified as two subsequent magnetic substorms.

Our database has a potential of adding valuable information to hitherto known global geomagnetic data resources. Recently, for instance, the HISTMAG database was created by Arneitz et al. (2017), consisting of large amounts of paleomagnetic and archeomagnetic data as well as a rich collection of instrumental historical observations. Compiling the historical data from
several observatories can be helpful in dealing with inconsistencies or missing data. The study of multiple records is also necessary for a more global view of a considered geophysical phenomenon, in contrast to the study of data from a single location on the Earth's surface. For example, in (Valach et al., 2019), the Clementinum data together with the geomagnetic records from other observatories have been used to analyse selected strong geomagnetic disturbances.

In a broader context, the scientific benefit of the presented work can be twofold. First, using the properly scaled series of
geomagnetic records, the long term behaviour of the geomagnetic field at the Clementinum observatory can be reconstructed. These data along with historical data from other observatories combined with some supplementary (archeomagnetic, paleomagnetic) data can be useful for modelling the global secular variation. Second, the consistent historical magnetic record supported with some other sources of data (e.g. records of aurorae) can be searched for extreme geomagnetic disturbances (magnetic storms and substorms). This kind of approach can make contribution to the study of extreme space weather events.

*Data availability.* The final digital data of magnetic declination and horizontal intensity given in physical units are available on the website https://doi.org/10.5194/angeo-2021-11-supplement.



*Author contributions.* PH devised the idea of the study, organized the scan of yearbooks and data digitazion, carried out primary data processing and transformation of declination to physical units, and checked the consistency of the final data. FV calculated/verified the temperature coefficients for the bifilar magnetometer in the years 1840–1854, participated in the transformation of the horizontal intensity to

physical units, and pointed to the event on 3 September 1839. MR helped with the introductory and concluding sections, amended the whole text, and checked the manuscript.

*Competing interests.* The authors declare that they have no conflict of interests.

*Acknowledgements.* PH thanks to Leif Swalgaard (Stanford University) who drew his attention to the old magnetic observatory data. The publication has been created thanks to the financial support of the Scientific Grant Agency of the Ministry of Education of Slovak Republic

and the Slovak Academy of Sciences, grant VEGA no. 2/0085/21.





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





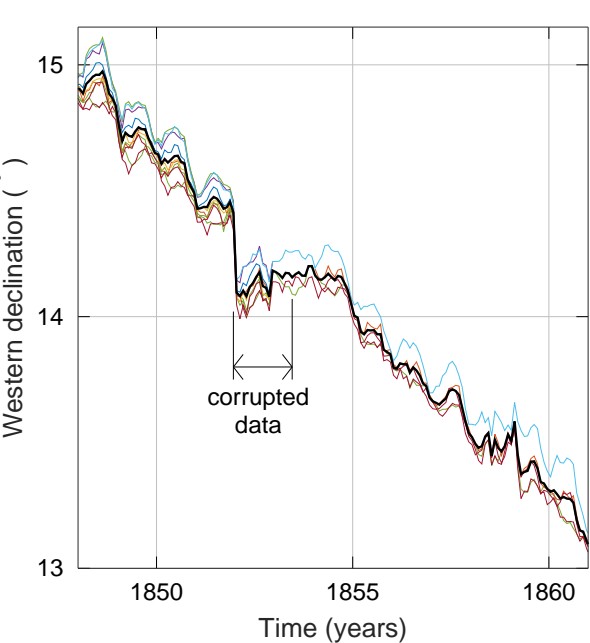

**Figure 1.** Time series of western declination according to observations at the Clementinum Observatory in Prague for the period from the beginning of 1848 to the end of 1860. The values measured at different parts of the days are plotted in different colours. The average daily values are shown by a thicker black curve. The period with unreliable data on variations of declination (which we discuss in Section 3) is pointed out.





**Figure 2.** Time series of monthly averages of declination (western), which was observed between 1850 and 1855 at the Clementinum Observatory in Prague at 10.00 pm. The yellow dots supplement the back-calculated results of the absolute measurements for 1852 at about the time of the summer solstice, the autumn equinox, and the winter solstice.



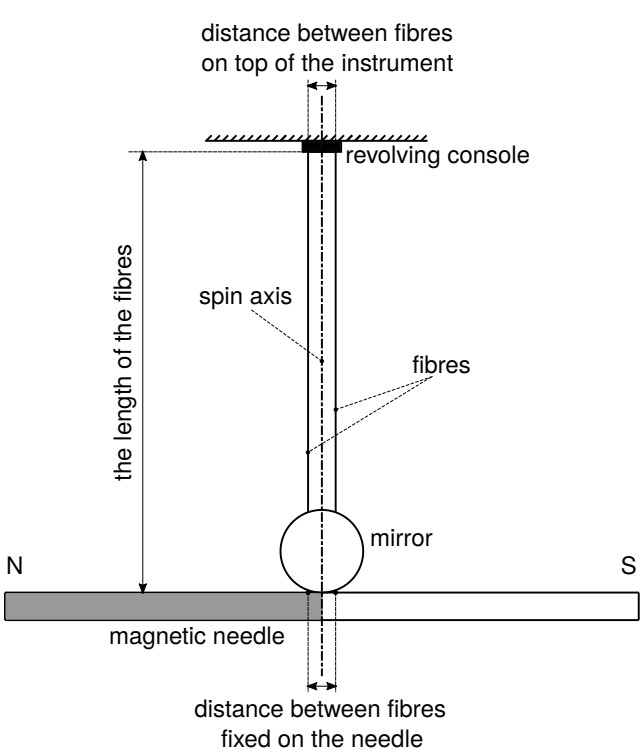

**Figure 3.** A sketch of the fibres, magnetic needle and mirror in the bifilar magnetometer.



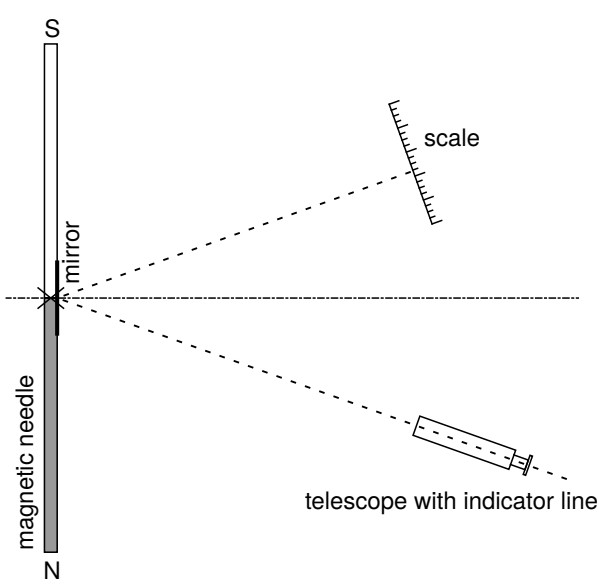

**Figure 4.** The top view of the bifilar magnetometer, mirror, telescope and scale.



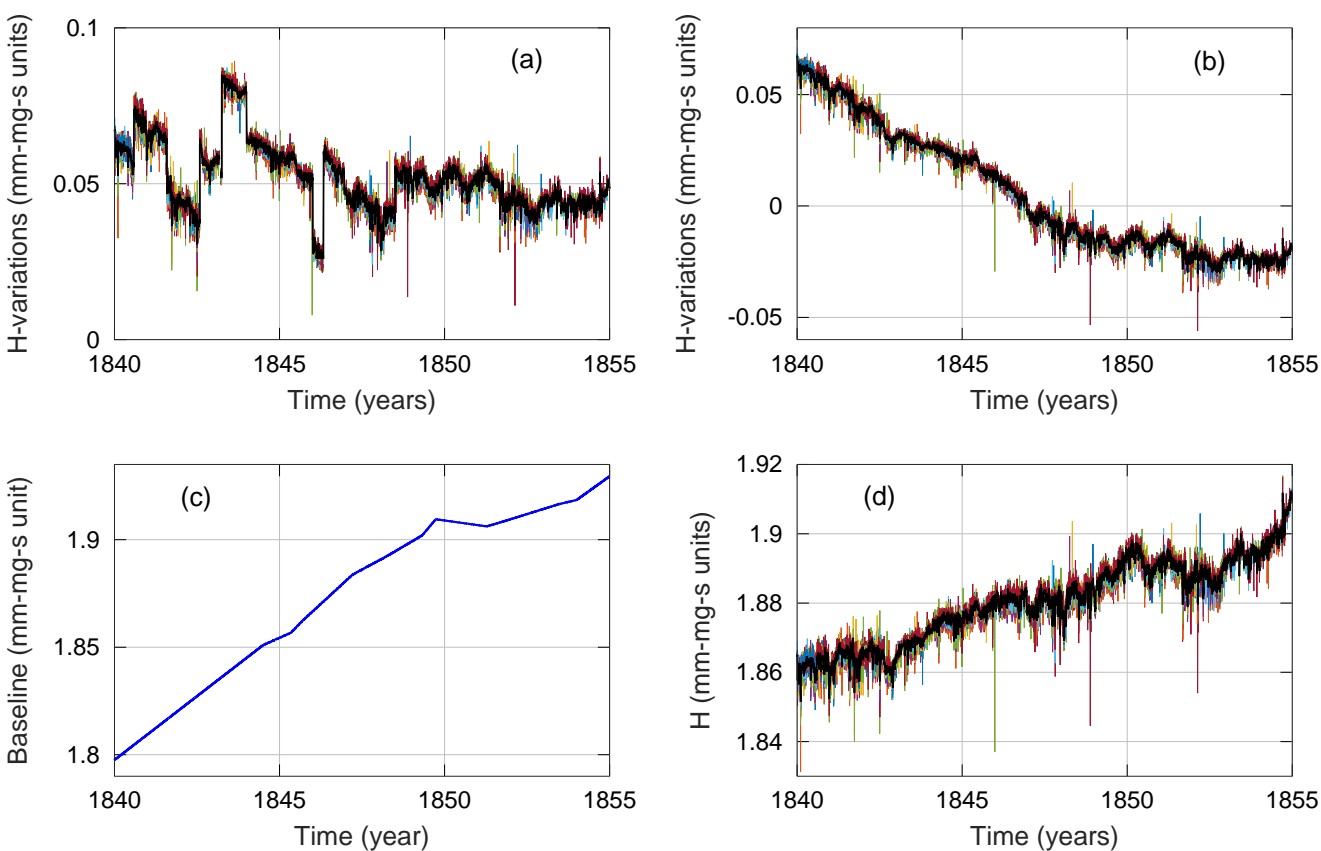

**Figure 5.** The horizontal intensity in Prague (Clementinum) from the beginning of 1840 until the end of 1854. (a) The variations as recorded in the yearbooks converted to the physical unit of the mm-mg-s system introduced by C. F. Gauss. Discontinuities that arose at the time when the settings of the bifilar apparatus were performed are not yet removed from the time series. The discontinuity between 1845 and 1846 arose from the replacement of the old apparatus with a new one. The values measured at different parts of the days are plotted in different colours. The average daily values are shown by a thicker black curve. (b) The time series after the removal of the discontinuities. (c) The baseline of the instrument that was used for measuring the variations of the horizontal intensity. (d) Time series of absolute values of the horizontal intensity at the Clementinum observatory in the period 1840–1854.

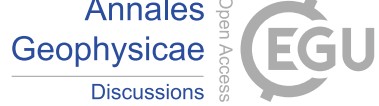

**Figure 6.** The comparison of the absolute values of the horizontal intensity observed at Clementinum (absolute measurements – blue colour, IGRF model – red curve, annual means calculated by us – black curve, the annual mean for 1855 taken from the yearbook – black asterisk) with the annual means of the horizontal intensity in Munich reduced by 600 nT (i.e., 0.06 absolute units mm-mg-s). The red curve is the horizontal intensity for Prague provided by the IGRF model.

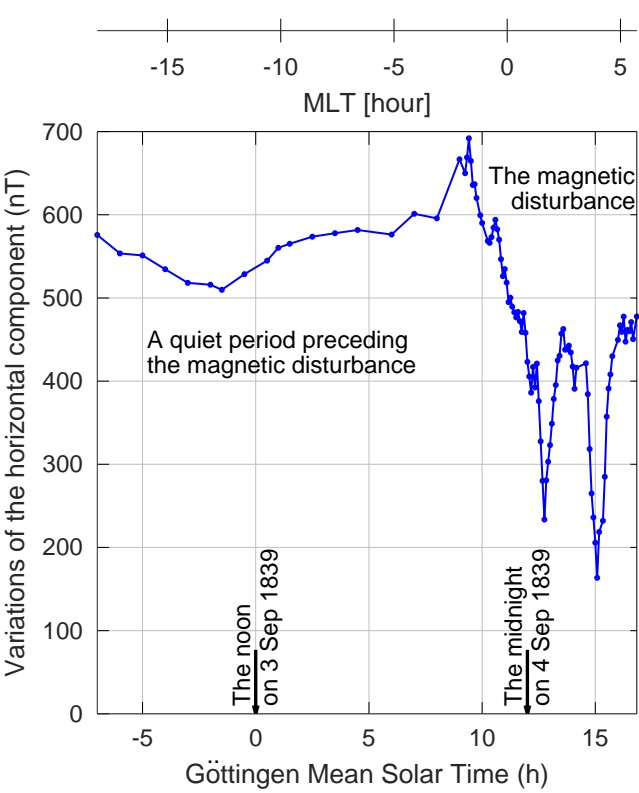

**Figure 7.** The first storm that was registered at Clementinum. It commenced on 3 September 1839 in the evening. Two sharp and deep depressions of the horizontal intensity occurred after midnight on 4 September 1839 at 12:45 h and 15:05 h of Göttingen mean solar time. These moments correspond to magnetic local times 01:39 MLT and 04:59 MLT, respectively. Judging from their shapes and from the fact that they happened at night, these two swift disturbances might probably have been substorms. The figure also demonstrates the procedure with which the magnetic storms were commonly registered that time: i.e. shortening the time interval between readings of the values from the variation instruments immediately after an interesting magnetic variation commenced, as described in Section 2.3.



**Table 1.** Summary of daily measurements published in the yearbooks Magnetische und meteorologische Beobachtungen zu Prag.

| Vol. | Years | Components | Time of measurements | Comments |
|---|---|---|---|---|
| 1 | Jul–Dec 1839 | D, H, I | 5, 6, 7, 8, 9, 10, 10:30, 11:30, 12:30, 13, 13:30, 14:30, 15:30, 16:30, 18, 19, 20, 21, 22 | scale units |
| 1 | Jan–Jul 1840 | D, H, I | 0, 2, 6, 7, 8, 9, 10, 11, 12, 13, 14, 15, 16, 17, 18, 19, 20, 21, 22, 23 | scale units |
| 2 | Aug–Dec 1840 | D, H, I | 0, 2 or 4, 6, 8, 10, 12, 13, 14, 16, 18, 20, 22 | scale units |
| 2 | Jan–Jul 1841 | D, H, I | 0, 4, 6, 8, 10, 12, 13, 14, 16, 18, 20, 22 | scale units |
| 3–4 | Aug 1841–Dec 1843 | D, $\Delta$D, H, $\Delta$H, I | 6, 8, 10, 12, 13, 14, 16, 18, 20, 22 | scale units |
| 5–6 | 1844–1845 | D, $\Delta$D, H, $\Delta$H, I | 6, 8, 10, 12, 13, 14, 16, 18, 20, 22 | scale units; not I at 22 h |
| 7–11 | 1846–Apr 1850 | D, $\Delta$D, H, $\Delta$H | 6, 8, 10, 12, 13, 14, 16, 18, 20, 22 | scale units |
| 11–13 | May 1850–Dec 1852 | D, H, $\Delta$H, I | 6, 8, 10, 12, 13, 14, 16, 18, 20, 22 | scale units |
| 14 | 1853 | D, H, I | 6, 14, 22 | scale units |
| 15–30 | 1854–1869 | D, H, I | 6, 8, 10, 14, 22 | scale units |
| 31–32 | 1870–1871 | D, H, I | 6, 10, 14, 18, 22 | scale units |
| 33–44 | 1872–1883 | D, H, I | 6, 10, 14, 18, 22 | scale units, D and H also in physical units |
| 45–50 | 1884–1889 | D, H | 6, 10, 14, 18, 22 | physical units |
| 51–53 | 1890–1892 | D, H | 6, 10, 14, 22 | physical units |
| 54 | 1893 | D, H | 6, 7, 14, 21 | physical units |
| 55–65 | 1894–1904 | D, H | 7, 14, 21 | physical units |
| 66–78 | 1905–1917 | D | 7, 14, 21 | physical units; increasing urban noise |

$\Delta$D = difference D(t) – D(t – 5 min), and similarly for $\Delta$H



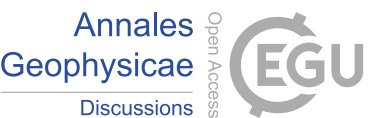

**Table 2.** The temperature coefficients found in the yearbook (Böhm and Karlinski, 1857, p. XV) and afterwards corrected by us.

| Year | $c_2$ (in parts of the whole horizontal intensity per °R) |
|------|-----------------------------------------------------------|
| 1846 | 0.001 172 |
| 1849 | 0.001 273 |
| 1856 | 0.001 108 |





**Table 3.** The values of the temperature coefficient $c_2$ for six continuous periods from the beginning of year 1840 to the end of year 1845.

| The beginning of the continual period | The end of the continual period | $c_2$ (in parts of the whole horizontal intensity per $^\circ$R) |
|:---:|:---:|:---:|
| 1/1/1840 | 31/7/1840 | 0.000 678 |
| 1/8/1840 | 1/8/1841 | 0.000 487 |
| 2/8/1841 | 31/7/1842 | 0.000 620 |
| 1/8/1842 | 31/3/1843 | 0.000 488 |
| 1/4/1843 | 31/12/1843 | 0.000 539 |
| 1/1/1844 | 31/12/1845 | 0.000 630 |



**Table 4.** The values of the temperature coefficient $c_2$ for continuous periods in years 1840-1845.

| The beginning of the continual period | The end of the continual period | $c_2$ (in parts of the whole horizontal intensity per $^\circ$R) |
| :---: | :---: | :---: |
| 1/1/1846 | 30/4/1846 | 0.001 340 |
| 1/5/1846 | 30/6/1848 | 0.000 932 |
| 1/7/1848 | 23/1/1849 | 0.001 102 |
| 24/1/1849 | 12/9/1854 | 0.001 262 |
| 13/9/1854 | 8/3/1855 | 0.001 292 |



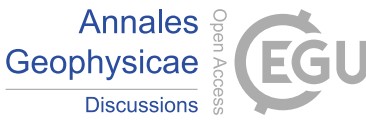

**Table 5.** Annual means of the horizontal intensity at the Clementinum observatory in Prague in the first years of its operation.

| Epoch (year) | Horizontal intensity (mm-mg-s) | (nT) |
|---|---|---|
| 1840.5 | 1.8620 | 18 620 |
| 1841.5 | 1.8643 | 18 643 |
| 1842.5 | 1.8641 | 18 641 |
| 1843.5 | 1.8683 | 18 683 |
| 1844.5 | 1.8746 | 18 746 |
| 1845.5 | 1.8779 | 18 779 |
| 1846.5 | 1.8810 | 18 810 |
| 1847.5 | 1.8800 | 18 800 |
| 1848.5 | 1.8820 | 18 820 |
| 1849.5 | 1.8882 | 18 882 |
| 1850.5 | 1.8919 | 18 919 |
| 1851.5 | 1.8901 | 18 901 |
| 1852.5 | 1.8877 | 18 877 |
| 1853.5 | 1.8937 | 18 937 |





**Table 6.** Geographical coordinates of Clementinum, Ashurst, and Dulwich Wood – another place which R. Snow used for his observations of the auroras that he reported about in (Snow, 1842); this place is situated in South London.

|  | Latitude | Longitude |
| --- | --- | --- |
| Clementinum | 50.08°N | 14.42°E |
| Ashurst | 51.27°N | 0.02°W |
| Dulwich Wood | 51.43°N | 0.01°W |