# Peer review of "The geomagnetic data of the Clementinum observatory in Prague since 1839"

_Annales Geophysicae, 2021_

## Author Response (AR1)

**Answers to Referee 1 comments**

*The authors are very grateful to the Referee for his/her valuable comments, which helped us to improve the quality of the manuscript.*

Review of the manuscript 'The geomagnetic data of the Clementinum observatory in Prague since 1839' by Pavel Hejda, Fridrich Valach and Milos Revallo, https://doi.org/10.5194/angeo-2021-11.

The manuscript describes a valuable and successful data recovery effort, to which the authors are to be congratulated, and makes a first interpretation of the obtained data. I strongly recommend publication of the manuscript after minor revisions.

General comments:
* * *
There are not many references and parts of the manuscript could be improved by adding earlier relevant studies. That would benefit the reader.

A number of references could be added that deal with historic geomagnetic time series in Europe but are not included in the introduction or discussion, e.g.:

Malin, S.R.C. and Bullard, E.C., 1981. The direction of Earth's magnetic field in London, 1570-1075. Phil. Trans. R. Soc. London, 299, 357-423

Alexandrescu, M., Courtillot, V., LeMouël, J.-L., 1996. Geomagnetic field direction in Paris since the mid-sixteenths century. Phys. Earth Planet. Inter., 98, 321-360

Korte, M., Mandea, M. and Matzka, J., 2009. A historical declination curve for Munich from different data sources. Phys. Earth Planet. Inter., 177, pp. 161-172, doi:10.1016/j.pepi.2009.08.005

Especially the last reference could be useful here as Munich is very close to Prague. The dataset presented in the manuscript would also lend itself to study historical Sq and the authors might mention this, cf.:

Cnossen, I. and Matzka, J., 2016. Changes in solar quiet magnetic variations since the Maunder Minimum: A comparison of historical observations and model simulations. J. Geophys. Res. Space Physics, 121, 10,520–10,535, doi:10.1002/2016JA023211

Referring to some review articles on geomagnetic observatories would also be helpful for the reader.

*Answer: We have involved new information together with corresponding references, which are as follows.*

*Lines 25-32: Arneitz et al., 2017a; Schröder and Wiederkehr, 2000; Reay et al., 2011; Matzka et al., 2010; Malin and Bullard, 1981; Alexandrescu et al., 1996; Korte et al., 2009; Pro et al., 2018*

*Lines 36-39: Lockwood et al., 2017, 2018; Cnossen and Matzka, 2016; Ptitsyna et al., 2018; Korte et al., 2009; Dobrica et al., 2018*

*Line 102: Wolf, 1859, 1860; Wolfer, 1914; Svalgaard, 2009, 2012*

I regard it as very important to also show the declination values (or any other component that was additionally measured) for the storm of 1939 in Figure 7.

*Answer: We have added declination to Figure 7 and a simple interpretation to Section 5 (Lines 434-440 of the new manuscript). We have also mentioned incomplete records of inclination. The caption of the figure has been completed.*

It would be nice to add plots of the data in the supplement, like annual means, all years, or daily means for each year. That would allow interested readers a quick evaluation of the data quality. (I have not plotted the data myself to evaluate the quality.)

*Answer: We have added plots of the data (monthly means) in the supplement.*

Detailed comments (number refers to line number)
* * *
**18**

supplied -> supplemented?

*Answer: Corrected to "supplemented". (Line 17 of the new manuscript.)*

**19**

to past few decades -> to the past few decades

*Answer: Corrected. (Line 18 of the new manuscript.)*

Note: The manuscript is well written, still it would profit from a native speaker quickly checking it. I refrain from further language corrections.

*Answer: We have striven for the grammatical correctness of the text. Besides, in e-mail communication with the journal editor, we were assured of that remaining language issues will be fixed by the publisher during the editing process.*

**23**

Please explain the term 'scale units'.

*Answer: We have added a short explanation of the term 'scale unit' in a parenthesis in Line 23.*

**28**

mention space climate

*Answer: We have mentioned space climate in Line 36 of the new manuscript.*

**47**

1936 -> 1836

*Answer: Corrected. (Line 59 of the new manuscript.)*

70

shine?

*Answer: "shine" was used as a synonym of "nice" (weather). The Referee was right indicating that the close relation to "sunny" could lead to misunderstanding. The word "shine" was therefore deleted. (Line 82 of the new manuscript.)*

**166**

Ernest -> Ernst

*Answer: Corrected. (Line 179 of the new manuscript.)*

**166**

Remove 'Out of GMU'

*Answer: Removed. (Line 179 of the new manuscript.)*

**193**

Mention in this paragraph that the next paragraph will explain how you determined the substitute values.

*Answer: The mention of the content of the following paragraphs has been added. (Line 208 of the new manuscript.)*

**208**

You estimate the annual mean from three measurements in the second half of the year. So your annual mean seems to be representative for the second half of the year, not for the full year. You

could further take into account the estimated secular variation to estimate the annual mean for the centre of the full year.

*Answer: We used the assumed uniform secular variation and information about the diurnal variation to estimate the annual mean for 1852. (Lines 220-225 of the new manuscript.)*

**212**

Same comment as I had for line 208.

*Answer: We used the assumed uniform secular variation to estimate the annual mean for 1853. (Lines 229-232 of the new manuscript.)*

**379 (and Figure 6)**

Alken et al., 2021, International Geomagnetic Reference Field: the thirteenth generation, states that IGRF covers 1900 to 2025, but you use it for the 1840ies and 1850ies. Please clarify.

*Answer: Thanks to the reviewer for notifying us of this error. In fact, we used the gufm1 model for the data in the 19th century. We used an online calculator available on the NCEI website. We have added this fact to the Acknowledgments. On that website, the data before 1900, i. e. model gufm1, can be obtained under the item marked as IGRF (1590-2024), which confused us. In the revised text, we changed the IGRF to gufm1 everywhere (namely: Line 386 and the legend and caption to Figure 6). In Line 386 and in caption to Figure 6, we have also added a reference to the article (Jackson et al., 2000), which presents the gufm1 model. In Acknowledgments we included the National Centers for Environmental Information (NCEI) for the operation of the online Magnetic Field Calculators and mentioned the web address of the calculator, which we used in our study.*

**Section 5, Figure 7**

Please add declination measurements to plot and discuss

*Answer: We have added declination to Figure 7 and a simple interpretation to Section 5 (Lines 434-440 of the new manuscript). We have also mentioned incomplete records of inclination. The caption of the figure has been completed.*

**407**

registration -> recording

*Answer: Corrected. (Line 445 of the new manuscript.)*

**Figure 1**

Please explain the colour of the plotted lines either in a legend or in the caption. Please also indicate the exact starting time of the corrupted data period.

*Answer: The plotted lines are explained in the new version of Figure 1. The exact starting time of the corrupted data period is now more apparent for we removed the connecting line during the jump (when the corrupted data started). The caption of the figure has been completed.*

**Figure 5 c**

Please indicated baselines determined by absolute measurements by symbols, then the reader can see on which data the blue line is based.

*Answer: Done.*

**Answers to Referee 2 comments**

Review of the manuscript "The geomagnetic data of the Clementinum observatory in Prague since 1839" by Pavel Hejda et al.

*The authors are very grateful to the Referee for his/her valuable and inspiring comments, which helped us to improve the quality of the manuscript.*

This manuscript presents a detailed analysis of historical geomagnetic measurements performed and recorded at the Clemetinum in Prague. The data set presents an unqiue and very valuable time series of such measurements starting in 1839. With great care the authors analyzed this data in order to reconstruct temporal variations of geomagnetic components in current physical units. Particularly the identification of possible error sources was treated with great care. Although the manuscript is very well written, there are some parts which definitely could be improved by proofreading of a native speaker. Nevertheless, I highly welcome this contribution and suggest acceptance of this manuscript after the following minor aspects have been considered.

*Answer concerning the proofreading: In e-mail communication with the journal editor, we were assured of that remaining language issues will be fixed by the publisher during the editing process.*

General remarks:

1) The significance of historical records for the analysis of recurrence rates of geomagnetic storms/disturbances could be discussed more prominently. In the past years estimations on recurrence rates, amplitudes and consequences of geomagnetic storms are gaining more and more interest. Such historical data sets, as analyzed in this study, are an important source for such statistical and periodicity analyses. Thus they are highly valuable when it comes to estimating the possible severity of upcoming space weather events. Such recurrence rates are discussed for example in the following articles:

Riley P., On the probability of occurrence of extreme space weather events. Space Weather, 2012

Love J.J., Credible occurrence probabilities for extreme geophysical events: Earthquakes, volcanic eruptions, magnetic storms. Geophysical Research Letters, 2012

*Answer: Thanks to the Referee for this inspiring comment.*

*Historical geomagnetic data sets do provide an important source for periodicity analysis of extreme geomagnetic events. To estimate recurrence rates, magnitudes and consequences of magnetic storms is crucial for understanding the possible severity of upcoming space weather events.*

*Unfortunately, observations of magnetic storms, such as the September 3, 1839 event, do not form a homogeneous time series in records from the Clementinum Observatory. The period when the*

*observers recorded the course of the storms in detail is limited to the first part of the observatory's operation, later the observers stopped this type of observation.*

*We have so far studied selected intensive geomagnetic storms from Clementinum, and in addition to the storm of September 1839, we have also examined the storms of November 1848 and February 1872. Including the Carrington storm of September 1859 (which has been actually failed to observe in Prague), we could infer a recurrence rate of extreme magnetic storms in the range of 9 to 13 years. On the other hand, the result in this form is in fact of little value because it does not take into account, for example, the variability of the level of geomagnetic activity in individual cycles of solar/geomagnetic activity.*

*Nonetheless, we are grateful to the Referee for this inspiration, as we will continue to study the space-weather aspects of Clementinum records (and also records from the other historical observatories) in our future work, while also looking for a satisfactory answer to the Referee's question.*

2) The discussion and description of the old instrumentation would further profit from some statements regarding the dynamic range of these instruments. As the authors have a profound knowledge on the physical limitations of the historical measurement systems, such discussion would definitely provide an added value at least to section 5. Insights on how strong geomagnetic variations have been during these events and whether the records can represent true amplitudes would be a nice addition.

*Answer: We added a new paragraph in Lines 419-433, where we provided a short discussion of limitations of the bifilar device.*

3) I wonder how a IGRF model can be obtained for the 19th century, as such models are only available since the 20th century. To my understanding, this is not possible. Please clarify what model you actually used. You are also comparing your data set to data from Munich. I did not find any citation, however. Please add a reference. (When posting my review I have seen that RC1 suggested that as well and provided a some important references)

*Answer: Thanks to the reviewer for notifying us of this error. In fact, we used the gufm1 model for the data in the 19th century. We used an online calculator available on the NCEI website. We have added this fact to the Acknowledgments. On that website, the data before 1900, i. e. model gufm1, can be obtained under the item marked as IGRF (1590-2024), which confused us. In the revised text, we changed the IGRF to gufm1 everywhere (namely: Line 386 and the legend and caption to Figure 6). In Line 386 and in caption to Figure 6, we have also added a reference to the article (Jackson et al., 2000), which presents the gufm1 model. In Acknowledgments we included the National Centers for Environmental Information (NCEI) for the operation of the online Magnetic Field Calculators and mentioned the web address of the calculator, which we used in our study.*

*The data of Munich were taken from the website of **World Data Centre of Geomagnetism (Edinburgh),** http://www.geomag.bgs.ac.uk/data_service/data/annual_means.shtml. The information added in Line 396. We have included this information also in Acknowledgments.*

Specific remarks:

*We thank the Referee for his/her specific remarks. We have accepted all of them in the new manuscript. Namely:*

5: by the then observers -> by the observers

*Line 5 of the new manuscript.*

47: in 1936 -> in 1836

*Line 59 of the new manuscript.*

52f: Joseph Stepling, started soon also  -> "Joseph Stepling, also started" or "started ... as well".

*Line 64-65 of the new manuscript.*

78: of lost by fire -> of loss by

*Line 89 of the new manuscript.*

100: citation for Wolf and Wolfer

*Reference added. (Line 102 of the new manuscript.)*

106: In Valach et al. (2019) …

*Line 119 of the new manuscript.*

120: Emperor Garden -> Imperial Garden

*Line 133 of the new manuscript.*

198: have already been mentioned

*Line 211 of the new manuscript.*

211: by the then value -> by the value

*Line 226 of the new manuscript.*

225: clarified -> described

*Line 242 of the new manuscript.*

227: used for to perform the -> used for performing observations

*Line 244 of the new manuscript.*

228: recenetly been reminded ... (1979) -> Well, I would just say: has been discussed by…

*Line 245 of the new manuscript.*

230: was -> is

*Line 247 of the new manuscript.*

232: and hanged close to -> better just say: and close to

*Line 248 of the new manuscript.*

362: by moving manually -> by manually moving

*Line 378 of the new manuscript.*

384: This benefit is related to records …

*Line 401 of the new manuscript.*

417: to that in Nevanlinna (1997).

*Line 455 of the new manuscript.*

427: in Valach et al. (2019)

*Line 465 of the new manuscript.*